# DC Charging Capabilities of Battery-Integrated Modular Multilevel Converters Based on Maximum Tractive Power

Arvind Balachandran [1,*], Tomas Jonsson [1,2] and Lars Eriksson [1]

1   Department of Electrical Engineering, Linköping University, SE-581 83 Linköping, Sweden
2   Scania AB, SE-151 48 Södertalje, Sweden
*   Correspondence: arvind.balachandran@liu.se

**Abstract:** The increase in the average global temperature is a consequence of high greenhouse gas emissions. Therefore, using alternative energy carriers that can replace fossil fuels, especially for automotive applications, is of high importance. Introducing more electronics into an automotive battery pack provides more precise control and increases the available energy from the pack. Battery-integrated modular multilevel converters (BI-MMCs) have high efficiency, improved controllability, and better fault isolation capability. However, integrating the battery and inverter influences the maximum DC charging power. Therefore, the DC charging capabilities of 5 3-phase BI-MMCs for a 40-ton commercial vehicle designed for a maximum tractive power of 400 kW was investigated. Two continuous DC charging scenarios are considered for two cases: the first considers the total number of submodules during traction, and the second increases the total number of submodules to ensure a maximum DC charging voltage of 1250 V. The investigation shows that both DC charging scenarios have similar maximum power between 1 and 3 MW. Altering the number of submodules increases the maximum DC charging power at the cost of increased losses.

**Keywords:** EV powertrain; DC charging; batteries; DC–AC converters; MMC; BI-MMC; AC batteries; reconfigurable batteries; modular batteries

## 1. Introduction

Over the last several decades, the average global temperature has risen considerably due to greenhouse gas emissions, and the automotive industry contributes about 15% of the emissions [1,2]. It is essential to increase the utilization of alternative energy carriers to replace fossil fuels. Automotive battery packs are typically made up of modules containing several parallel and/or series-connected cells [3]. However, the energy and power are determined not only by the cell type and size but, to a large extent, also by the configuration and battery management system (BMS) [4,5]. By restructuring the cell interconnections and introducing more electronics in the pack, more precise control and, thus, better utilization of the energy in the individual modules can increase the energy and provide more benefits such as improved battery life and increased usable capacity of the battery pack [6,7].

Currently, EV powertrains typically utilize a large battery pack with a conventional two-level voltage source inverter [8]. The battery pack typically contains low-voltage battery cells (e.g., 2–4 V) connected in parallel to achieve the required power rating. These cells are then connected in series, providing high-voltage (e.g., 300–1000 V) [9]. Because of differences in leakage currents and cells in homogeneity, individual cell voltage and state-of-charge (SOC) distribution among the cells are non-homogeneous. As a result, some cells discharge faster than other cells, thus limiting the total energy the pack can deliver. Cell balancers are employed as part of the battery management systems (BMS) to mitigate this problem [4]. However, individual cell control is desirable to maximize the energy delivered by the battery pack. This is achieved by integrating power electronics into the battery pack, thereby changing the battery interconnection pattern in response to

the battery behavior and user demands. This provides enhanced fault tolerance, charge and temperature balancing, extended energy delivery, and easy integration of batteries of different ages and chemistry types [10].

Modular multilevel converters (MMCs) have gained popularity in the power distribution sector, especially in HV and MV applications where it has been proven to give several advantages, such as low THD, high modularity, and scalability [11,12]. Furthermore, over the last few years, battery-integrated MMCs (BI-MMCs) have gained popularity in battery energy storage systems (BESS) [13–15]. References [16,17] indicate a significant benefit in increasing the controllability of cells in terms of battery lifetime and battery utilization. A slight increase in the battery lifetime and utilization typically results in tremendous benefits [18]. BI-MMCs are, thus, particularly interesting for EV powertrains because of their high efficiency, greater cell-level control, and provide better battery fault isolation [19–25]. Low power (7.4 to 43 kW) AC and higher power (63 to 350 kW) DC charging capabilities for cascaded H-bridge topologies are presented in [26,27]. These articles report several advantages with BI-MMCs while charging, such as active balancing during charging, flexible DC charging voltage, and the potential elimination of a dedicated onboard charger for AC charging. Although the shown interesting effort in the literature, the mega-watt (MW) DC charging capabilities of BI-MMCs were not investigated.

The charging time for electric vehicles is significantly longer than the refueling time for conventional vehicles. To achieve a short charging time, efficient DC fast chargers capable of delivering high power are required. As a result, different standards for DC fast-charging systems are developed [28]. The combined charging system (CCS) is a standard for charging electric vehicles and can provide power up to 350 kW [29–31]. A key challenge in electrifying heavy-duty vehicles (HDVs) such as 40-ton commercial vehicles, is the need for high-energy storage capacity [32]. The anticipated size of the battery packs for HDVs is about 250 to 750 kWh [31]. To meet the changing needs of medium- and heavy-duty commercial vehicles' large energy storage system's short charging time intervals of 30 to 40 min, megawatt charging systems (MCS) are under development [33]. MCS chargers have an estimated charging power of 1 MW or greater with a maximum charging voltage and current of 1250 V and 3000 A, respectively [34].

The battery pack is connected directly to the fast charger in a conventional powertrain. However, the battery and the inverter are integrated into a BI-MMC, potentially increasing the DC fast charging capabilities because higher voltages are achieved during charging than during traction.

*Contributions and Outline*

The first contribution involves the derivation of the maximum DC charging power of five three-phase BI-MMCs, considering the same submodule semiconductor losses for a maximum tractive power of 400 kW for a 40-ton commercial vehicle. The second contribution is a comparative assessment of five three-phase BI-MMCs with 1, 6, and 12 cascaded cells per submodule, considering two different design criteria either based on the maximum motor voltage or maximum MCS DC charger voltage. The assessment includes the maximum DC charging power, voltage, and current, the total number of submodules, submodule losses, total semiconductor losses, and submodule temperature at maximum charging power.

The article outline follows: Section 2 presents an overview of the five BI-MMC topologies. Section 3 presents the two different design criteria to determine the total number of submodules either based on the maximum motor voltage or maximum MCS DC charger voltage. Section 4 describes the power loss calculations for the two different design criteria. The maximum DC charging current and power calculations are described in Section 5. Section 6 presents the calculations of the submodule case temperature at maximum charging power. Section 7 presents the comparative assessment of 5 3-phase BI-MMCs with 1-, 6-, and 12-cascaded cells per submodule. Finally, discussions and conclusions are presented in Sections 8 and 9, respectively.

## 2. Topology Review

Figure 1 presents the schematic of BI-MMC topologies. They consist of either one or two arms per phase ($N_{arms}$), and each arm is made up of several cascaded stages of power converters and is commonly referred to as submodules (SM) ($N_{sm,arm}$ SMs per arm). In the figure, the terminals 'P' and 'N' are used as the positive and negative terminals for DC charging, and the circuit breaker, $\overline{CB_n}$, in the open position ensures that the electric machine (EM) is disconnected from the BI-MMC during DC charging. Figure 1a,b present the double-star half-bridge (DSHB) and double-star full-bridge (DSFB) topologies, respectively, and Figure 1c,d gives the single-star half-bridge and single-star full-bridge topologies, respectively. In single-star topologies, in addition to $\overline{CB_n}$ open, it is also necessary to ensure that $CB_n$ is open for DC charging. In double-star BI-MMCs, arm inductors are used to reduce the amplitude of circulating currents. Still, in single-star topologies, there is no path for the circulating current during traction. Therefore, arm inductors are not required for such a design. Figure 1e illustrates the single-delta topology. In this topology, in addition to $\overline{CB_n}$ open, $CB_p$ and $CB_n$ should be in position 'Y' for DC charging. A detailed description of all the topologies is presented in [35]. The SMs are bidirectional by design due to the anti-parallel diode, and as a result, the AC side current can be controlled in both directions.

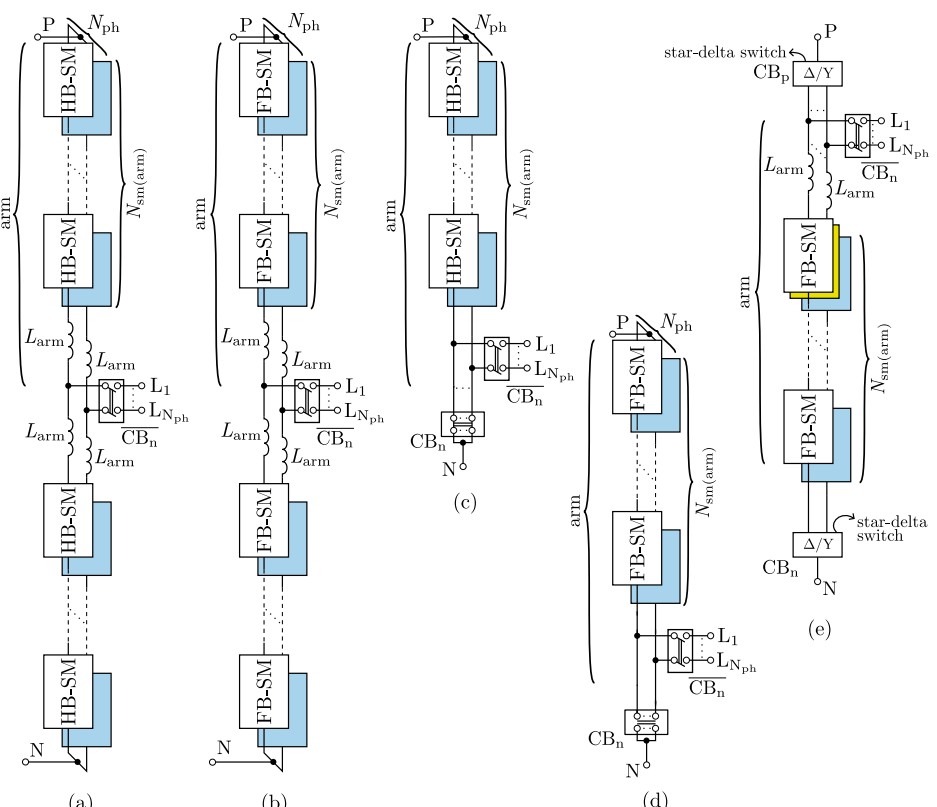

**Figure 1.** Schematic of battery-integrated modular multilevel converters (BI-MMCs) for an $N_{ph}$-phase system during DC charging. (**a**) Double-star half-bridge (DSHB), (**b**) double-star full-bridge, (**c**) single-star half-bridge (SSHB), (**d**) single-star full-bridge (SSFB), and (**e**) single-delta full-bridge (SDFB) topologies [36].

Figure 2a shows the schematic of a typical megawatt (MW) DC charger. A medium voltage (MV) three-phase electrical grid is connected to an active rectifier (AC/DC) and followed by a stage of DC-to-DC converter (typically, a dual-active bridge) [37–39]. The output of the DC-to-DC converter stage is connected to the 'P' and 'N' terminals of the BI-MMC through the MCS connector for DC charging [40]. Figure 2b shows the constant current (CC) and constant current constant voltage (CC-CV) charging process. A detailed description of the charging process is described in [4]. The DC charger controls the current

through the BI-MMC during DC charging. The figure clearly shows that the charging power varies throughout the entire charging cycle. However, the maximum DC charging power ($P_{max}^c$), such that the semiconductor losses per submodule during the traction and charging are equal, is presented in Section 5.

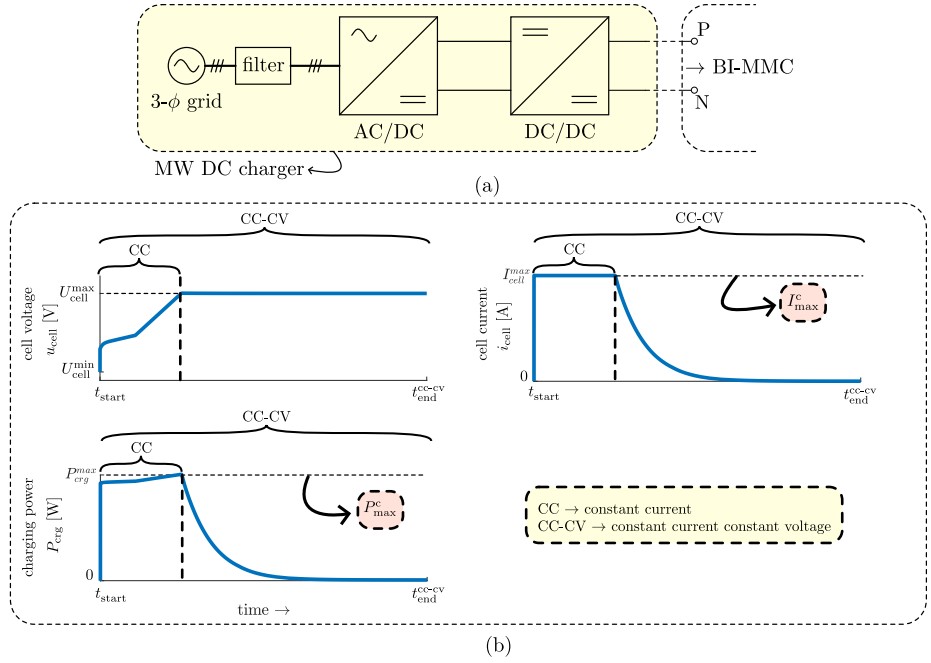

**Figure 2.** Schematic of a megawatt DC charger and different charging strategies. (**a**) megawatt Dc charger schematic and (**b**) the constant current constant voltage (CC-CV) and constant current (CC) charging strategies.

Figure 3a,b present the half-bridge (HB) and full-bridge (FB) submodules (SM), respectively. The figure shows that the DC side of an SM contains a battery pack, configured with $N_{s(cells)}$ series and $N_{p(cells)}$ parallel cells, and DC-link capacitors modeled as an RLC circuit with an equivalent series resistance (*ESR*), equivalent capacitance, (*C*) and parasitic inductance between the capacitors and the high-side switches (*ESL*). $N_{s(cells)}$ defines the desired SM DC voltage ($U_s$), and the required battery capacity per submodule defines $N_{p(cells)}$. An SM consists of 2 or 4 switches (for HB- and FB–SMs, respectively), and each switch is made of $N_{p(mos)}$ parallel MOSFETs. The HB-SM, shown in Figure 3a, has two complementary switches $S_1$ and $S_2$. When $S_1$ is 'off' ($S_2$ is 'on'), $u_{sm}$ is equal to 0 V, referred to as the bypass state. Alternatively, when $S_1$ is 'on' ($S_2$ is 'off'), the SM output voltage $u_{sm}$ is equal to the DC side voltage $U_s$; this is referred to as the insertion state. The FB–SM, shown in Figure 3b, has four switches, $S_1$, $S_2$, and $S_3$, $S_4$, where $S_1$, $S_2$, and $S_3$, $S_4$ are complementary switches. When either $S_1$, $S_3$, or $S_2$, $S_4$ is 'on', $u_{sm}$ is 0 V (bypass states). When $S_1$ and $S_4$ are 'on' ($S_2$ and $S_3$ are 'off'), then $u_{sm}$ is equal to $U_s$ (insertion state). Similarly, when $S_2$ and $S_3$ are 'on' ($S_1$ and $S_4$ are 'off'), then $u_{sm}$ is equal to $-U_s$ (insertion state). The RMS output voltage of the HB-SM ($U_{sm(hb)}$) and FB–SM ($U_{sm(fb)}$) are:

$$U_{sm(hb)} = M_{max}\frac{U_s}{2\sqrt{2}}, \qquad\qquad U_{sm(fb)} = M_{max}\frac{U_s}{\sqrt{2}}, \qquad (1)$$

where $M_{max}$ is the maximum modulation index.

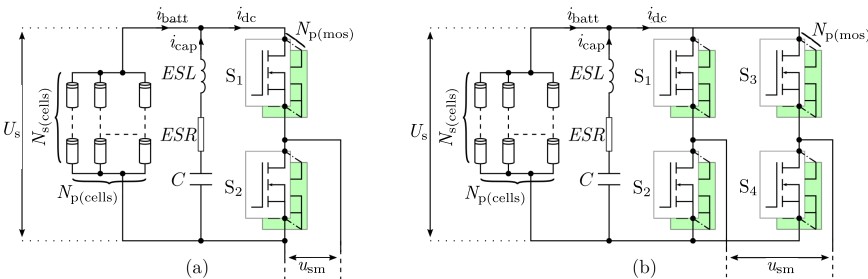

**Figure 3.** Schematic of battery-integrated MMC submodules. (**a**) half-bridge submodule (HB-SM) and (**b**) full-bridge submodule (FB–SM) [36].

## 3. Total Number of Submodules

This section presents the two different methods of determining the total number of submodules for continuous DC charging: CDC-T gives the total number of submodules determined by the traction voltage, and CDC-C presents the total number of submodules by the maximum DC charger voltage.

### 3.1. CDC-T: Total Number of Submodules Determined by the Traction Voltage

During traction, the SMs are operated as DC–AC inverters and the total number of submodules ($N_{\text{sm}}^{\text{t}}$) required to achieve an output RMS phase-to-neutral voltage of $U_{\text{ph}}$ is calculated using the following relation:

$$N_{\text{sm}}^{\text{t}} = \frac{U_{\text{ph}}}{U_{sm}} N_{\text{arms}} N_{\text{ph}}, \tag{2}$$

where $N_{\text{ph}}$ is the number of phases.

### 3.2. CDC-C: Total Number of Submodules Determined by Maximum DC Charger Voltage

During DC charging, the SMs are used as DC–DC buck converters and the BI-MMC DC-terminal voltage ($U_{\text{pn}}$) is given as follows:

$$U_{\text{pn}} = \frac{N_{\text{sm}}^{\text{t}}}{N_{\text{ph}}} N_{\text{arms}} U_{\text{s}}, \tag{3}$$

One way to maximize the DC charging power is to ensure that $U_{\text{pn}}$ is equal to the maximum voltage of the DC charger ($U_{\text{dc(c)}}^{\text{max}}$) and the total number of submodules required to ensure $U_{\text{pn}} = U_{\text{dc(c)}}^{\text{max}}$ ($N_{\text{sm}}^{\text{c}\star}$) is calculated as follows:

$$N_{\text{sm}}^{\text{c}\star} = \frac{U_{\text{dc(c)}}^{\text{max}}}{U_{\text{s}} N_{\text{arms}}} N_{\text{ph}}. \tag{4}$$

If a BI-MMC topology has $N_{\text{sm}}^{\text{c}\star}$ SMs, resulting in a phase-to-neutral RMS AC output voltage of $U_{\text{ph}}'$, and $N_{\text{sm}}^{\text{c}\star} < N_{\text{sm}}^{\text{t}}$, then $U_{\text{ph}}' < U_{\text{ph}}$. As a result, the BI-MMC cannot reach the maximum traction voltage, reducing traction power. Therefore, the total number of submodules ($N_{\text{sm}}^{\text{c}}$) required to ensure $U_{\text{pn}} > U_{\text{dc(c)}}^{\text{max}}$ while also ensuring a maximum AC output voltage of $U_{\text{ph}}$ is determined as follows:

$$N_{\text{sm}}^{\text{c}} = \max\left(N_{\text{sm}}^{\text{t}}, N_{\text{sm}}^{\text{c}\star}\right). \tag{5}$$

It is important to mention that when the total number of submodules ($N_{sm}$) is altered, the total number of parallel cells per SM will also change. This is because the total energy stored in the batteries is the same. As a result, during charging, the change in the total number of SM batteries in series compensates for the change in the number of parallel cells per SM. Therefore, the battery losses in both CDC-T and CDC-C are identical.

## 4. Power Loss Calculations

This section presents the power loss calculations during traction and DC charging.

### 4.1. Power Loss during Traction

The maximum arm current during traction ($I_{arm}^t$) is calculated as follows:

$$I_{arm}^t = \frac{P_{max}^t}{N_{ph}\, U_{ph}\, \cos(\phi)\, N_{arms}},$$

(6)

where $\cos(\phi)$ is the traction motor power factor and $P_{max}^t$ is the maximum tractive power.

The conduction and switching losses of a switch ($P_{c,sw}^{l(t)}$ and $P_{s,sw}^{l(t)}$, respectively) are determined as follows:

$$P_{c,sw}^{l(t)} = \frac{1}{2}\left(I_{arm}^t\right)^2 \frac{R_{ds(on)}}{N_{p(mos)}}, \qquad P_{s,sw}^{l(t)} = \frac{2\sqrt{2}}{\pi}\, U_s\, I_{arm}^t\, t_{sw(tran)} f_{sw}^t,$$

(7)

where $R_{ds(on)}$ is the MOSFET on-state resistance, $N_{p(mos)}$ is the number of parallel MOSFETs per switch, $t_{sw(tran)}$ is the combined switching transient time, corresponding to the sum of current rise and voltage fall time at turn-on and the voltage rise and current fall time at a turn-off, i.e., $t_{sw(tran)} = t_{ri} + t_{fi} + t_{rv} + t_{fv}$, and $f_{sw}^t$ is the MOSFET switching frequency. $N_{p(mos)}$ is calculated, considered a maximum case temperature, $t_{sw(tran)}$ is determined considering a maximum drain-to-source voltage ripple, and $f_{sw}^t$ is selected such that the DC-current harmonic components are bypassed by the DC-link capacitors [36].

The total losses in a submodule during traction ($P_{sm}^{l(t)}$) are given as follows:

$$P_{sm}^{l(t)} = \left(P_{c,sw}^{l(t)} + P_{s,sw}^{l(t)}\right) N_{sw},$$

(8)

where $N_{sw}$ represents the number of switches per SM. It is important to mention that the SM circuit board contains the switches and the DC-link capacitors. As a result, the total losses per submodule include both $P_{sm}^{l(t)}$ and the capacitor losses per SM. However, due to the design choice of the DC-link capacitors, the capacitor losses per SM are far lower than $P_{sm}^{l(t)}$ [36]. Therefore, the total losses per submodule are equal to $P_{sm}^{l(t)}$.

The total semiconductor losses during traction ($P_{sc}^{l(t)}$) is given as follows:

$$P_{sc}^{l(t)} = P_{sm}^{l(t)} N_{sm(tot)},$$

(9)

where $N_{sm(tot)}$ represents the total number of submodules presented in either CDC-A or -B.

### 4.2. Power Loss during DC Charging

As mentioned previously, during DC charging, the SMs of the BI-MMCs are operated as DC–DC buck converters, and the SM duty cycles ($D_c$) are equal to 1, i.e., SMs are always inserted. However, for the topologies where $U_{pn} > U_{dc(c)}^{max}$, then $D_c = U_{pn}/U_{dc(c)}^{max}$. It is worth mentioning that $D_c$ among SMs can be different and is determined by a BMS active balancing algorithm to ensure an even SOC distribution among the SM cells. Furthermore, the DC charging current magnitude is determined by the charger, and it is assumed that there exists communication between the vehicle and the DC charger to control the charging current.

The distribution of losses within the SM depends on $D_c$, i.e., during the insertion period; $S_1$ in HB-SM, and $S_1$ and $S_3$ in the FB–SM, bare the conduction losses; and during the bypass period, the other switches bare the conduction losses. The DC charging conduction losses per switch during the insertion- and bypass-states ($P_{c,sw(ins)}^{l(c)}$ and $P_{c,sw(byp)}^{l(c)}$, respectively) are given as follows:

$$P^{l(c)}_{c,sw(ins)} = D_c (I^c_{arm})^2 \frac{R_{ds(on)}}{N_{p(mos)}}, \qquad P^{l(c)}_{c,sw(byp)} = (1 - D_c)(I^c_{arm})^2 \frac{R_{ds(on)}}{N_{p(mos)}}, \qquad (10)$$

where $I^c_{arm}$ is the DC arm current during charging and $R_{ds(on)}$ is the MOSFET on-state resistance.

In the continuous DC charging (CDC) case, the MOSFET switching frequency is equivalent to the rate of active balancing determined by the BMS, and the switching losses are neglected. The total losses in a submodule during DC charging, $P^c_{sm}$, is, thus, given as follows:

$$P^c_{sm} = \left( P^{l(c)}_{c,sw(ins)} + P^{l(c)}_{c,sw(byp)} \right) \frac{N_{sw}}{2}. \qquad (11)$$

The total semiconductor losses during DC charging ($P^{l(c)}_{sc}$) is given as follows:

$$P^{l(c)}_{sc} = P^c_{sm} N_{sm(tot)}. \qquad (12)$$

## 5. Maximum DC Charging Power Calculations

In a conventional powertrain, during DC charging, the positive and negative terminals of the battery pack are connected to the DC charger, and the losses incurred are only in the battery. However, in a BI-MMC-based powertrain, the battery and the inverter are integrated, and as a result, the losses during DC charging are increased. Therefore, to restrict the losses and cooling requirements per submodule, the submodule losses per charging and traction are considered to be equal, i.e.,

$$P^{l(t)}_{sm} = P^c_{sm}, \qquad \Longrightarrow P^{l(t)}_{sm} = \left( P^{l(c)}_{c,sw(ins)} + P^{l(c)}_{c,sw(byp)} \right) \frac{N_{sw}}{2}. \qquad (13)$$

The maximum DC charging arm current to ensure that the total semiconductor charging and traction losses are equal ($I^{c,max}_{arm}$) can, thus, be calculated as follows:

$$I^{c,max}_{arm} \approx \sqrt{\frac{2 P^{l(t)max}_{sm} N_{p(mos)}}{R^{max}_{ds(on)} N_{sw}}}, \qquad (14)$$

where $P^{l(t)max}_{sm}$ is the SM losses at $P^t_{max}$ and $R^{max}_{ds(on)}$ is the MOSFET on-state resistance at maximum junction temperature.

The maximum DC charging power is calculated using the following:

$$P^c_{max} = U^{max}_{dc(c)} I^c_{max}, \qquad \text{where } I^c_{max} = N_{ph} I^{c,max}_{arm}. \qquad (15)$$

## 6. Submodule Case Temperature

The SM case temperature ($T_c$) is calculated using the following relation:

$$T_c = R_{\theta ca} P^l_{sm} + T_a, \qquad (16)$$

where $R_{\theta ca}$ is the case of ambient thermal resistance (presented in Appendix A), $P^l_{sm}$ is the submodule losses, and $T_a$ is the ambient temperature.

## 7. Comparative Assessment

The BI-MMC design parameters are presented in Table 1. The converter design considers a maximum tractive power of 400 kW and a 20-pole traction motor with a nominal speed of 1000 rpm. A maximum modulation index ($M_{max}$) of 0.85 was considered, allowing for 15% redundant submodules; 24 Ah Samsung NMC Li-ion cells were considered with nominal and minimum cell voltages of 3.7 V and 3.45 V, respectively. The minimum cell voltage selected from the open circuit voltage vs. state-of-charge curve corresponds to 65% depth-of-discharge. The total energy stored in the batteries of a 40-ton commercial

vehicle is assumed to be one MWh. Appendix A shows the number of parallel MOSFETs per switch, the maximum drain-to-source resistances, the MOSFET switching frequencies, and the case of ambient thermal resistance, determined using the procedure shown in [36].

**Table 1.** Design parameters for a 400 kW 40-ton commercial vehicle.

| Parameters | Symbol | Value |
|---|---|---|
| Maximum tractive power | $P_{max}^t$ | 400 kW |
| AC phase-to-phase voltage | $U_v$ | 440 V |
| Electric machine nominal speed | - | 1000 rpm |
| Load power factor | $\cos(\phi)$ | 0.9 |
| Maximum modulation index | $M_{max}$ | 0.85 |
| MCS DC charging voltage [†] | $U_{dc(c)}^{max}$ | 1250 V |
| MCS DC charging current [†] | $I_{mcs}^c$ | 3000 A |
| MOSFET CDC switching frequency | CDC–$f_{sw}^c$ | ≈1 mHz |
| Total energy stored in the batteries | $E_{batt}$ | 1 MWh |

[†] MCS standards [34].

The two different DC charging scenarios for the comparative assessment are as follows:

CDC-T Continuous DC charging with the total number of submodules determined by the traction voltage.

CDC-C Continuous DC charging with the total number of submodules determined by the maximum DC charger voltage.

### 7.1. Number of Submodules

Figure 4 shows the total number of submodules determined by the traction voltage ($N_{sm}^t$) and maximum DC charger voltage ($N_{sm}^{c\star}$) for all BI-MMC topologies with 1, 6, and 12 $N_{s(cells)}$.

### 7.1.1. $N_{s(cells)}$ Comparison

The figure clearly shows that the total number of submodules (both $N_{sm}^t$ and $N_{sm}^c$) decreases with an increase in $N_{s(cells)}$ for a given topology. This is because as $N_{s(cells)}$ increase, the DC-side SM voltage ($U_s$) increases, thus increasing the SM output RMS voltage ($U_{sm}$), and this, in turn, reduces the total number of submodules required to have $U_{ph}$ ($U_{ph}$ is the same for all topologies and $N_{s(cells)}$).

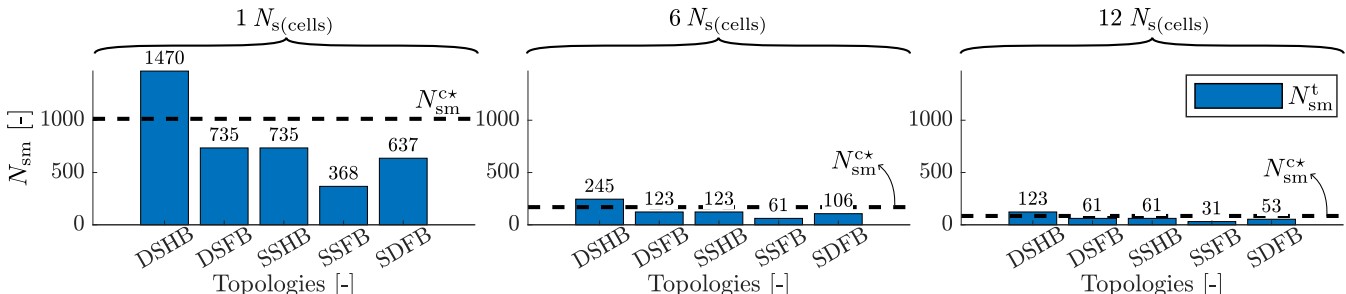

**Figure 4.** Total number of submodules determined by the traction voltage ($N_{sm}^t$) and maximum DC charger voltage ($N_{sm}^{c\star}$) for all topologies with 1, 6, and 12 $N_{s(cells)}$.

### 7.1.2. Topology Comparison

From the figure, it is clear that DSHB has a 50% lower $N_{sm}^t$ than DSFB. This is because $U_{sm}$ for DSFB is two times more than that of DSHB because of the bi-polar nature of FB–SMs. For the same reason, SSFB has 50% lower $N_{sm}^t$ than SSHB, and $N_{sm}^t$ for DSFB and SSHB are identical for a given $N_{s(cells)}$. SDFB has $\sqrt{3}$ times higher $N_{sm}^t$ than that of SSFB because, in the SDFB, $U_v$ and $U_{ph}$ are equal.

$N_{\mathrm{sm}}^{\mathrm{c}\star}$ for all topologies is identical by the definition of CDC-C. However, in the DSHB topology, $N_{\mathrm{sm}}^{\mathrm{t}}$ is greater than $N_{\mathrm{sm}}^{\mathrm{c}\star}$ since $U_{\mathrm{pn}}$ is greater than $U_{\mathrm{dc(c)}}^{\mathrm{max}}$. Consequently, the maximum AC traction phase-to-neutral voltage for DSHB with $N_{\mathrm{sm}}^{\mathrm{c}\star}$ submodules is lower than $U_{\mathrm{ph}}$, thus resulting in lower tractive power. Therefore, in CDC-C, DSHB $N_{\mathrm{sm}}^{\mathrm{c}}$ and $N_{\mathrm{sm}}^{\mathrm{t}}$ are the same, and during DC charging, the $D_{\mathrm{c}}$ of DSHB is equal to $U_{\mathrm{dc(c)}}^{\mathrm{max}}/U_{\mathrm{pn}}$. $N_{\mathrm{sm}}^{\mathrm{c}}$ for all other topologies is the same as $N_{\mathrm{sm}}^{\mathrm{c}\star}$.

### 7.2. Submodule Losses

Figure 5 presents the submodule semiconductor losses ($P_{\mathrm{sm}}^{\mathrm{l}}$) for the two different DC charging cases, namely, CDC-T and CDC-C at a maximum charging power of $P_{\mathrm{max}}^{\mathrm{c}}$ for all BI-MMC topologies with 1, 6, and 12 $N_{\mathrm{s(cells)}}$. $P_{\mathrm{sm}}^{\mathrm{l}}$ for both the DC charging scenarios is identical, and this is, by definition, i.e., ensuring that the submodule losses during charging and traction are identical.

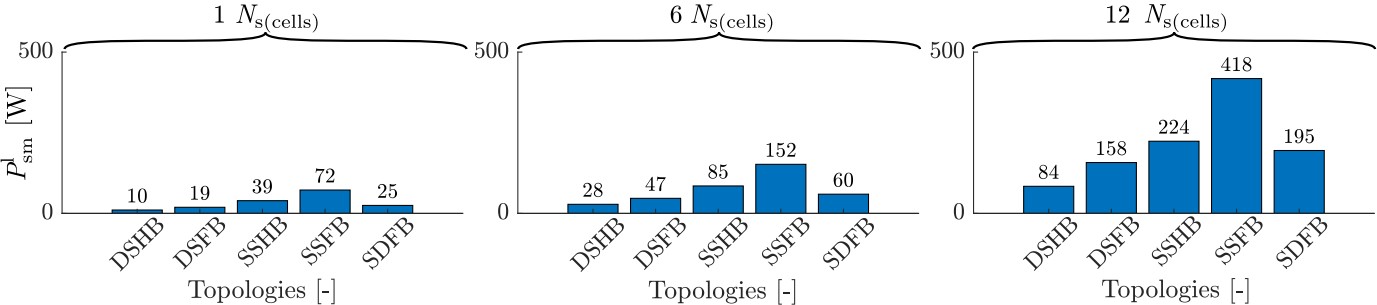

**Figure 5.** Total submodule losses for all the DC charging scenarios considering a maximum charging power of $P_{\mathrm{max}}^{\mathrm{c}}$ for all topologies with 1, 6, and 12 $N_{\mathrm{s(cells)}}$.

#### 7.2.1. $N_{\mathrm{s(cells)}}$ Comparison

It is clear that $P_{\mathrm{sm}}^{\mathrm{l}}$ increases with an increase in $N_{\mathrm{s(cells)}}$ for a given topology. This is because of the increase in the conduction losses due to the high $R_{\mathrm{ds(on)}}$ of the higher voltage class MOSFETs employed at higher $N_{\mathrm{s(cells)}}$.

#### 7.2.2. Topology Comparison

The DSFB has about two times more $P_{\mathrm{sm}}^{\mathrm{l}}$ than the DSHB because the DSFB has two times more $N_{\mathrm{sw}}$ than the DSHB for a given $N_{\mathrm{s(cells)}}$. For the same reason, $P_{\mathrm{sm}}^{\mathrm{l}}$ for SSFB is two times more than in SSHB. $P_{\mathrm{sm}}^{\mathrm{l}}$ for SSHB is almost four times as in DSHB because SSHB has two times more $I_{arm}$ than DSHB. For the same reason, SSFB has three times more $P_{\mathrm{sm}}^{\mathrm{l}}$ than DSFB. For 12 $N_{\mathrm{s(cells)}}$, $P_{\mathrm{sm}}^{\mathrm{l}}$ of SSHB is a factor of 3 higher than DSHB because SSHB has slightly higher $N_{\mathrm{p(mos)}}$ than DSHB, and a detailed calculation for $N_{\mathrm{p(mos)}}$ is described in [36]. For the same reason, SSFB has three times more $P_{\mathrm{sm}}^{\mathrm{l}}$ than DSFB at 12 $N_{\mathrm{s(cells)}}$. $P_{\mathrm{sm}}^{\mathrm{l}}$ for SSFB is about three times higher than in SDFB because $I_{arm}$ for SSFB is $\sqrt{3}$ times greater than in SDFB.

The SSFB has the highest $P_{\mathrm{sm}}^{\mathrm{l}}$ compared with the other topologies, but the thermal resistance of the SSFB submodule is relatively low (as shown in Table A1). As a result, the case temperature is kept under a maximum allowable case temperature ($T_{\mathrm{c}}^{\mathrm{max}}$) of 80 °C. (as shown in Section 7.6). Since $P_{\mathrm{sm}}^{\mathrm{l}}$ for FB–SM is two times that of the HB-SM, the cost of the cooling system for the FB–SMs is higher than HB-SMs. This is reflected in the case-to-ambient thermal resistance in Table A1.

### 7.3. Total Semiconductor Losses

Figure 6 presents the total semiconductor losses considering both $N_{\mathrm{sm}}^{\mathrm{t}}$ ($P_{\mathrm{sc}}^{\mathrm{l(t)}}$) (CDC-T) and $N_{\mathrm{sm}}^{\mathrm{c}}$ ($P_{\mathrm{sc}}^{\mathrm{l(c)}}$) (CDC-C) during traction considering a maximum power of 400 kW for all topologies with 1, 6, and 12 $N_{\mathrm{s(cells)}}$.

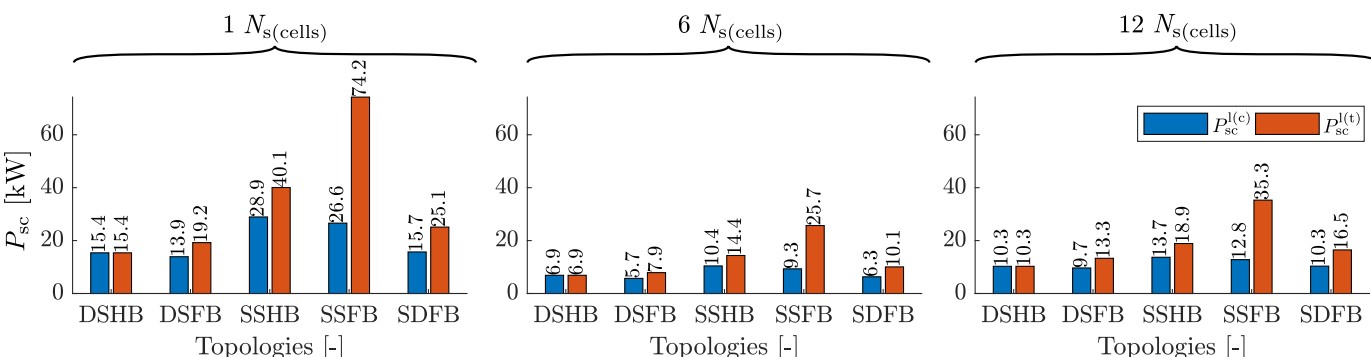

**Figure 6.** Total semiconductor losses considering both $N_{sm}^t$ ($P_{sc}^{l(t)}$), CDC-T, and $N_{sm}^c$ submodules ($P_{sc}^{l(c)}$), CDC-C, during traction at a maximum power of 400 kW for all topologies with 1, 6, and 12 $N_{s(cells)}$.

### 7.3.1. $N_{s(cells)}$ Comparison

For a given topology, the total semiconductor losses, $P_{sc}^l$ (both $P_{sc}^{l(t)}$ and $P_{sc}^{l(c)}$), are the lowest at 6 $N_{s(cells)}$. This is because as $N_{s(cells)}$ increases, the MOSFET $R_{ds(on)}$ increases but not in proportion to the total number of submodules (both $N_{sm}^t$ and $N_{sm}^c$) decreases.

### 7.3.2. Topology Comparison

For a given $N_{s(cells)}$ and topology: the losses per submodule for both the cases (CDC-C and CDC-T) are identical (by definition). As a result, $P_{sc}^{l(t)}$ and $P_{sc}^{l(c)}$ are proportional to $N_{sm}^t$ and $N_{sm}^c$, respectively. Therefore, all topologies except DSHB have higher $P_{sc}^{l(c)}$ than $P_{sc}^{l(t)}$ for a given $N_{s(cells)}$. In DSHB, $N_{sm}^c$ and $N_{sm}^t$ are the same; thus, $P_{sc}^{l(t)}$ and $P_{sc}^{l(c)}$ are equal. $P_{sc}^{l(c)}$ (CDC-C) for SSFB is about three times as $P_{sc}^{l(t)}$ (CDC-T) because $N_{sm}^c$ is about three times as $N_{sm}^t$.

$P_{sc}^{l(t)}$ for SSHB is about two times more than DSHB. This is because the arm current during traction ($I_{arm}^t$) for SSHB is two times more than DSHB, and $N_{sm}^t$ for DSHB is half as much as DSHB. For the same reason, $P_{sc}^{l(t)}$ for SSFB is two times more than DSFB. The SSFB has about $\sqrt{3}$ times higher $P_{sc}^{l(t)}$ than SDFB. This is because the $I_{arm}^t$ is $\sqrt{3}$ times higher and $N_{sm}^t$ is about a factor $\sqrt{3}$ lower in SSFB than SDFB. $P_{sc}^{l(t)}$ for DSHB and DSFB are almost identical. This is because the $N_{sm}^t$ for DSFB is half of DSHB, but DSFB has twice the number of switches as DSHB. For the same reason, $P_{sc}^{l(t)}$ for SSFB and SSHB are similar.

SSFB has about four times higher $P_{sc}^{l(c)}$ than DSHB. This is because the arm's current during charging ($I_{arm}^c$) is twice as much for SSFB than DSHB, and both topologies have identical $N_{sm}^c$. SSFB has about two times the $P_{sc}^{l(c)}$ as SSHB because $P_{sm}^l$ for SSFB is around twice as much as SSHB, and both topologies have identical $N_{sm}^c$. $P_{sc}^{l(c)}$ for DSFB is about 30% more than DSHB because DSFB has twice the $P_{sm}^l$ as DSHB, but $N_{sm}^c$ for DSHB is higher than in DSFB.

Although the SSFB CDC-C has about three times higher $P_{sc}^{l(c)}$ than the SSFB CDC-T, the SSFB CDC-C SM case temperature is lower than 80 °C. However, the high $P_{sc}^{l(c)}$ of SSFB CDC-C significantly increases the cost of cooling systems.

### 7.4. Maximum DC Charging Voltage and Current

Figure 7 shows the maximum BI-MMC DC link voltage and maximum DC charging current considering the two different scenarios, CDC-T and CDC-C, for all topologies with 1, 6, and 12 $N_{s(cells)}$. Figure 7a gives the maximum BI-MMC DC link voltage ($U_{pn}$) and maximum MCS DC charger voltage ($U_{dc(c)}^{max}$). The figure shows that $U_{dc(c)}^{max}$ is independent of $N_{s(cells)}$ for a given topology. This is because in CDC-T, the $N_{sm}^t$ is designed such that all topologies have the same $U_{ph}$, irrespective of $N_{s(cells)}$, and in CDC-C, $N_{sm}^c$ is determined such that $U_{pn}$ is equal to $U_{dc(c)}^{max}$, irrespective of $N_{s(cells)}$. $U_{pn}$ for DSHB in CDC-T and CDC-C are identical because both $N_{sm}^c$ and $N_{sm}^t$ for DSHB are equal. In CDC-T, the distribution of

$U_{\mathrm{pn}}$ among topologies follows $N_{\mathrm{sm}}^{\mathrm{t}}$ for a given $N_{\mathrm{s(cells)}}$. However, in CDC-C, by definition, $U_{\mathrm{pn}}$ and $U_{\mathrm{dc(c)}}^{\max}$ are equal for all topologies except DSHB. $U_{\mathrm{pn}}$ for DSHB is higher than $U_{\mathrm{dc(c)}}^{\max}$ because $N_{\mathrm{sm}}^{\mathrm{c}}$ is greater than $N_{\mathrm{sm}}^{\mathrm{c\star}}$.

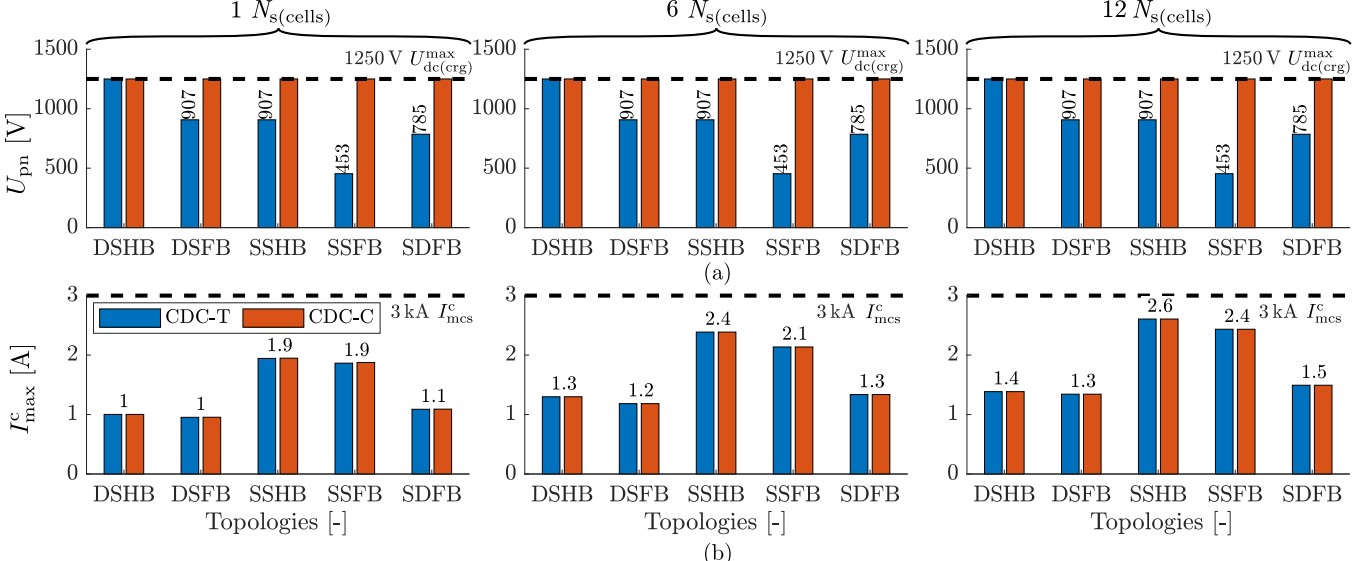

(a)

(b)

**Figure 7.** The maximum BI-MMC DC link voltage and maximum DC charger current considering the two different DC charging scenarios: CDC-T and CDC-C for all topologies with 1, 6, and 12 $N_{\mathrm{s(cells)}}$ with the maximum allowable DC voltage and current for MCS [34]. (**a**) maximum BI-MMC DC link voltage ($U_{\mathrm{pn}}$) and the maximum MCS DC charger voltage ($U_{\mathrm{dc(c)}}^{\max}$), and (**b**) maximum DC charger current ($I_{\max}^{\mathrm{c}}$) the maximum MCS DC charger current ($I_{\mathrm{mcs}}^{\mathrm{c}}$).

Figure 7b shows the maximum DC charging current ($I_{\max}^{\mathrm{c}}$), and it is clear that as $N_{\mathrm{s(cells)}}$ increases, $I_{\max}^{\mathrm{c}}$ increases marginally for a given topology. This is because $f_{\mathrm{sw}}^{\mathrm{t}}$ increases with the increase in $N_{\mathrm{s(cells)}}$. $I_{\max}^{\mathrm{c}}$ for CDC-T and CDC-C are similar for a given topology and $N_{\mathrm{s(cells)}}$, because $P_{\mathrm{sm}}^{\mathrm{l}}$ for CDC-T and CDC-C are similar (by definition). $I_{\max}^{\mathrm{c}}$ for DSHB and DSFB are similar even though $P_{\mathrm{sm}}^{\mathrm{l}}$ for DSFB is two times more than DSHB. This is because FB–SMs have twice the $N_{\mathrm{sw}}$ as HB-SMs, for a given $N_{\mathrm{s(cells)}}$. For the same reason, $I_{\max}^{\mathrm{c}}$ for SSHB and SSFB are similar for 1 $N_{\mathrm{s(cells)}}$. At 6 and 12 $N_{\mathrm{s(cells)}}$, however, $I_{\max}^{\mathrm{c}}$ SSFB is slightly lower than SSHB because these topologies have different $N_{\mathrm{p(mos)}}$. $I_{\max}^{\mathrm{c}}$ for SSFB is about two times more than in DSFB for a given $N_{\mathrm{s(cells)}}$. This is because $P_{\mathrm{sm}}^{\mathrm{l}}$ for SSFB is about four times more than in DSFB. For the same reason, $I_{\max}^{\mathrm{c}}$ for SSHB is about twice as DSFB. DSFB and SDFB have similar $I_{\max}^{\mathrm{c}}$ because these topologies have similar $P_{\mathrm{sm}}^{\mathrm{l}}$.

## 7.5. Maximum DC Charging Power

Figure 8 shows the maximum DC charging power ($P_{\max}^{\mathrm{c}}$) considering the two different DC charging scenarios, CDC-T and CDC-C, for all topologies with 1, 6, and 12 $N_{\mathrm{s(cells)}}$.

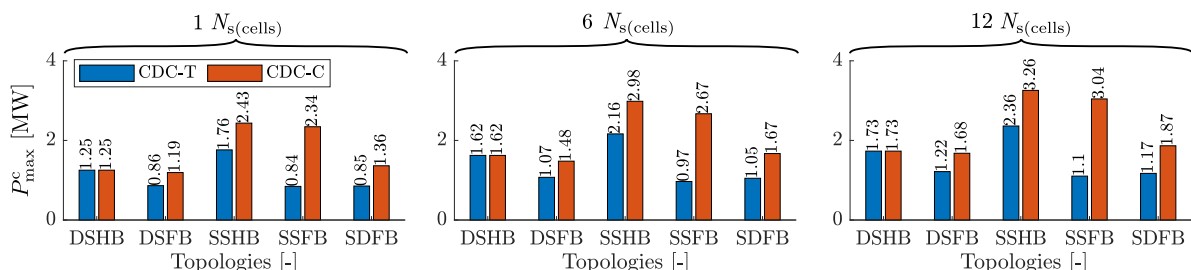

**Figure 8.** The maximum DC charging power considering the two different DC charging scenarios, CDC-T and CDC-C, for all topologies with 1, 6, and 12 $N_{\mathrm{s(cells)}}$.

### 7.5.1. $N_{s(cells)}$ Comparison

The figure shows that for a given topology, as $N_{s(cells)}$ increases, $P_{max}^c$ increases marginally. This is because $I_{max}^c$ increases marginally with an increase in $N_{s(cells)}$.

### 7.5.2. Topology Comparison

The figure shows that $P_{max}^c$ for CDC-C is higher than in CDC-T for all topologies except DSHB. This is because $U_{pn}$ in CDC-C is much greater than in CDC-T for all topologies except DSHB for a given $N_{s(cells)}$. In CDC-C, $P_{max}^c$ for all topologies follows $I_{max}^c$ for a given $N_{s(cells)}$ because $U_{pn}$ for all the topologies is the same.

In CDC-T, $P_{max}^c$ for SSFB and DSFB are similar for a given $N_{s(cells)}$. This is because $U_{pn}$ for SSFB is half of that in DSFB, but $I_{max}^c$ for SSFB is two times more than in DSFB. For the same reason, $P_{max}^c$ for SSHB is similar to that in DSHB. DSFB and SDFB have similar $P_{max}^c$ because these topologies have similar $U_{pn}$ and $I_{max}^c$, irrespective of the DC charging scenario (CDC-T or CDC-C).

$P_{max}^c$ for DSHB in CDC-T and CDC-C are identical because $N_{sm}^t$ and $N_{sm}^c$ are equal for a given $N_{s(cells)}$. $P_{max}^c$ for SSFB in CDC-C is about three times greater than in CDC-T because $N_{sm}^c$ is about three times higher than $N_{sm}^t$. For the same reason, DSFB, SSHB, and SDFB also have higher $P_{max}^c$ in CDC-C than in CDC-T and is proportional to the difference between $N_{sm}^c$ and $N_{sm}^t$.

All the BI-MMC topologies have a maximum DC charging power between 800 kW to 3.3 MW. This corresponds to a maximum charging C-rate between 1 C to 3 C assuming a 1 MWh battery system.

### 7.6. Submodule Temperature

$N_{p(mos)}$ is selected such that $T_c$ for all topologies is below 80 °C considering an ambient temperature of 40 °C, and is presented in Table A1. A minimum limit for $N_{p(mos)}$ of 4 is chosen to reduce the total losses. Figure 9 shows the $T_c$ for all topologies with 1, 6, and 12 $N_{s(cells)}$ at a maximum charging power of $P_{max}^c$.

### 7.6.1. $N_{s(cells)}$ Comparison

As $N_{s(cells)}$ increases, $T_c$ also increases. This is because the MOSFET on-state resistance also increases with an increase in $N_{s(cells)}$ (Table A1), thereby increasing $P_{sm}^l$.

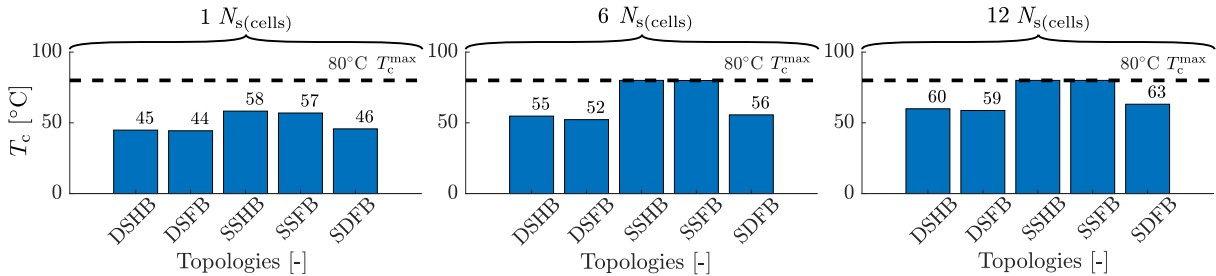

**Figure 9.** The submodule temperature for all topologies with 1, 6, and 12 $N_{s(cells)}$ at a maximum charging power of $P_{max}^c$.

### 7.6.2. Topologies Comparison

The double-star topologies have a lower $T_c$ than for a given $N_{s(cells)}$. This is because, in the double-star topologies, the RMS output current is split equally between the two arms resulting in lower losses. The SDFB is slightly more than in DSFB because $P_{sm}^l$ for SDFB is slightly more than in DSFB are similar. The $T_c$ for DSFB and DSHB are similar. This is because $P_{sm}^l$ for DSFB is twice as in DSHB, but DSFB has 50% lower $R_{\theta ca}$ than DSHB (see Table A1) since DSFB has twice as many switches as DSHB.

## 8. Discussion

The DC charging power can be increased for both CDC-T and CDC-C scenarios by increasing the maximum SM temperature above 80 °C during charging. However, this increases the total semiconductor losses.

The total submodule losses include all the switches in the SM. Therefore, during DC charging, the distribution of power losses among the switches within the SM is not even and is dependent on the duty cycle of the submodule.

The underlying assumption for the analysis is that the total semiconductor losses during charging and traction are identical. However, the vehicle is stationary during charging, which affects the cooling. Therefore, to ensure that total semiconductor losses during charging and traction are the same, possibly additional cooling requirements are required. If all the topologies had the same number of parallel MOSFETs, then submodule conduction losses during traction for the SSFB and SSHB topology would increase. This also increases the submodule conduction losses during DC charging. As a result, the total DC charging power will also increase, and so will the case temperature.

Extending the battery losses during traction from [36] to DC charging with a power of 1 MW and a DC-link voltage of 800 V, the battery losses are 4.5 kW. Furthermore, assuming that the total energy stored in the battery pack of the two-level inverter-based powertrain and the batteries in BI-MMCs is identical, the battery losses during charging for both powertrains are equal. However, the total losses during charging in a BI-MMC include the semiconductor losses much greater than the two-level inverter. Therefore, the total losses during charging in BI-MMCs are much higher than in a two-level inverter-based powertrain.

## 9. Conclusions

Two different DC charging scenarios for five different three-phase BI-MMC topologies with 1, 6, and 12 cascaded cells per submodule designed for a maximum tractive power of 400 kW for a 40-ton commercial vehicle were investigated. The two DC charging scenarios are continuous DC charging with the total number of submodules determined by the traction voltage (CDC-T) and continuous DC charging with the total number of submodules determined by the maximum DC charger voltage (CDC-C). A topology's maximum charging power ($P_{\max}^c$) is defined as the power at which the total semiconductor losses during traction and DC charging are equal.

Most BI-MMCs with the total number of submodules determined by the maximum DC charger voltage (CDC-C) have higher $P_{\max}^c$ than BI-MMCs with the total number of submodules determined by the traction voltage (CDC-T). In particular, SSFB $P_{\max}^c$ is about three times as high in CDC-C than in CDC-T. However, the total semiconductor losses ($P_{sc}^l$) are significantly higher in CDC-C than in CDC-T. As a result, the total power converter efficiency reduces, potentially reducing the advantages of BI-MMCs, especially during traction. For the DSHB, $P_{\max}^c$ in both CDC-C and CDC-T are identical. Therefore, $P_{\max}^c$ can be further increased at the cost of increased submodule losses.

About 20% of BI-MMC topologies with 6 and 12 $N_{s(cells)}$ have about 2.5 to 3.3 MW of $P_{\max}^c$, about 30% of all the topologies with 1, 6, and 12 $N_{s(cells)}$ have $P_{\max}^c$ of about 1.5 to 2.5 MW and all the other topologies have $P_{\max}^c$ of about 800 kW to 1.5 MW. All the BI-MMC topologies can achieve 1 h or shorter charging time, corresponding to 1 C or higher charging current.

**Author Contributions:** Conceptualization, A.B., T.J. and L.E.; methodology, A.B., T.J. and L.E.; software, A.B.; validation, A.B., T.J. and L.E.; formal analysis, A.B.; investigation, A.B.; resources, A.B.; data curation, A.B.; writing—original draft preparation, A.B.; writing—review and editing, A.B., T.J. and L.E.; visualization, A.B.; supervision, T.J. and L.E.; project administration, A.B., T.J. and L.E.; funding acquisition, T.J. and L.E. All authors have read and agreed to the published version of the manuscript.

**Funding:** This article is a part of the BattVolt project in the Mistra Innovation 23 research program, a research program financed by the Foundation for Strategic Environmental Research (MISTRA).

**Data Availability Statement:** Data supporting reported results can be found at https://gitlab.liu.se/BI-MMC_public/dc-charging-of-bi-mmcs/continuous-dc-charging (accessed on 23 January 2023).

**Conflicts of Interest:** The authors declare no conflict of interest.

**Appendix A**

Table A1 presents the number of parallel MOSFETs per switch ($N_{p(mos)}$) for all topologies at different $N_{s(cells)}$, such that the case temperature does not exceed 80 °C calculated using the relation in [36]. Most topologies have four $N_{p(mos)}$ because the minimum number of parallel MOSFETs is limited to 4. The table also presents the maximum on-state resistance ($R_{ds(on)}^{max}$) for all topologies at different $N_{s(cells)}$, and for a given $N_{s(cells)}$, all topologies employ the same MOSFET. Furthermore, the table presents the MOSFET switching frequency during traction ($f_{sw}^{t}$) for all topologies at different $N_{s(cells)}$ using the optimization principle presented in [36]. Finally, the table shows the SM case of ambient thermal resistance.

**Table A1.** The total number of parallel MOSFETs per switch, maximum MOSFET on-state resistance, MOSFET switching frequency during traction, and SM case-to-ambient thermal resistance for all topologies with 1, 6, and 12 $N_{s(cells)}$.

| Topology / $N_{s(cells)}$ | 1 | 6 | 12 |
|---|---|---|---|
| Total number of parallel MOSFETs ($N_{p(mos)}$) | | | |
| DSHB | 4 | 4 | 4 |
| DSFB | 4 | 4 | 4 |
| SSHB | 4 | 5 | 6 |
| SSFB | 4 | 4 | 5 |
| SDFB | 4 | 4 | 4 |
| Maximum MOSFET on-state resistance ($R_{ds(on)}^{max}$) | | | |
| - | 0.375 mΩ | 0.6 mΩ | 1.6 mΩ |
| MOSFET switching frequency during traction ($f_{sw}^{t}$) | | | |
| DSHB | 4.2 kHz | 7.5 kHz | 9.8 kHz |
| DSFB | 2.5 kHz | 5.2 kHz | 8.8 kHz |
| SSHB | 3.2 kHz | 5.5 kHz | 7.5 kHz |
| SSFB | 1.8 kHz | 3.5 kHz | 6.2 kHz |
| SDFB | 2.2 kHz | 4.8 kHz | 8.2 kHz |
| SM case-to-ambient thermal resistance ($R_{\theta ca}$) | | | |
| DSHB | 0.46 K/W | 0.52 K/W | 0.24 K/W |
| DSFB | 0.23 K/W | 0.27 K/W | 0.12 K/W |
| SSHB | 0.46 K/W | 0.52 K/W | 0.24 K/W |
| SSFB | 0.23 K/W | 0.27 K/W | 0.12 K/W |
| SDFB | 0.23 K/W | 0.27 K/W | 0.12 K/W |

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
