# Peer review of "DC Charging Capabilities of Battery-Integrated Modular Multilevel Converters Based on Maximum Tractive Power"

_electricity, doi:10.3390/electricity4010005_

Round 1

Reviewer 1 Report

-In Fig. 1, the arm inductors are not needed for single-star arrangements, since there is no ring current that can be formed. The load itself is inductive, and the additional arm inductance would only limit the permissible maximum power (only valid for AC).

- Only variables should be written in italic! This goes also for subscripts and exponents. For example, when you write Usm(hb), sm(hb) must be non-italic since it is not a variable. This should be corrected throughout the paper in all equations, figures and tables.

- Regarding Figs. 4 and 5, the losses of the SSFB submodules, is quite high. Is such an arrangement even feasible?

- The DSHB uses much more submodules and thus, much more switches are used. If you would use the same amount of switches (parallel) for the SSFB, how would the losses compare then? Can you make a small comment on that where you just adapt the conduction losses based on the number of switches?

-Can you compare the charging efficiencies of the different systems with a classical two-level inverter battery system when using DC charging? I guess the efficiency is much worse since the charging current is going through the switches of the submodules. A small calculation or a comparison with literature values should be sufficient.

Author Response

Thank you for reading the paper and providing your valuable comments. A figure is added in response to the comments from reviewer 2. Therefore, the figure references in the review comments are altered, referring to the figure numbers of the modified manuscript. 
I have attached the pdf file with a point-by-point response to the comments.

Reviewer 2 Report

Add More details about figure 1, show more details about the DC source.  

Show more details the control of the charging process, constant current or constant voltage.

Add more justifications for the need to use the proposed charging system

Show if the proposed system uni-directional or bi-directional

 Add more comments for the results in Figures 4 and 5.

If you have a heat sink for each topology this will add an additional cost but connect that to the complexity and reliability this will make your research more attractive. 

Author Response

Thank you for reading the paper and providing your valuable comments. Apologies that the manuscript is very dense and challenging to read. An English language check is performed on the manuscript using the typing assistant tool Grammarly, where the spelling, grammar, punctuation, clarity, engagement, and delivery are checked. 
A figure is added in response to the comments. Therefore, the figure references in the review comments are altered, referring to the figure numbers of the modified manuscript. 

I have attached the pdf file with a point-by-point response to the comments.

Round 2

Reviewer 2 Report

Yes I agree to publish the last version of the manuscript. 

Author Response

Thank you for reading the paper and providing your valuable comments.